# Compositional Optimization of Sputtered SnO_2_/ZnO Films for High Coloration Efficiency

**DOI:** 10.3390/ijms251910801

**Published:** 2024-10-08

**Authors:** Zoltán Lábadi, Noor Taha Ismaeel, Péter Petrik, Miklós Fried

**Affiliations:** 1Centre for Energy Research, Institute of Technical Physics & Materials Science, Konkoly-Thege Rd. 29-33, 1121 Budapest, Hungary; labadi.zoltan@ek.hun-ren.hu (Z.L.); noor.t@ilps.uobaghdad.edu.iq (N.T.I.); petrik.peter@ek.hun-ren.hu (P.P.); 2Doctoral School on Materials Sciences and Technologies, Óbuda University, 1034 Budapest, Hungary; 3Institute of Laser for Postgraduate Studies, University of Baghdad, Baghdad 10070, Iraq; 4Department of Electrical and Electronic Engineering, Institute of Physics, Faculty of Science and Technology, University of Debrecen, 4032 Debrecen, Hungary; 5Institute of Microelectronics and Technology, Óbuda University, Tavaszmezo Str. 17, 1084 Budapest, Hungary

**Keywords:** combinatorial sample, zinc-tin oxide, reactive sputtering, electrochromic materials, coloration efficiency

## Abstract

We performed an electrochromic investigation to optimize the composition of reactive magnetron-sputtered mixed layers of zinc oxide and tin oxide (ZnO-SnO_2_). Deposition experiments were conducted as a combinatorial material synthesis approach. The binary system for the samples of SnO_2_-ZnO represented the full composition range. The coloration efficiency (CE) was determined for the mixed oxide films with the simultaneous measurement of layer transmittance, in a conventional three-electrode configuration, and an electric current was applied by using organic propylene carbonate electrolyte cells. The optical parameters and composition were measured and mapped by using spectroscopic ellipsometry (SE). Scanning Electron Microscopy (SEM) and Energy-Dispersive X-ray Spectroscopy (EDS) measurements were carried out to check the SE results, for (TiO_2_-SnO_2_). Pure metal targets were placed separately from each other, and the indium–tin-oxide (ITO)-covered glass samples and Si-probes on a glass holder were moved under the two separated targets (Zn and Sn) in a reactive argon–oxygen (Ar-O_2_) gas mixture. This combinatorial process ensured that all the compositions (from 0 to 100%) were achieved in the same sputtering chamber after one sputtering preparation cycle. The CE data evaluated from the electro-optical measurements plotted against the composition displayed a characteristic maximum at around 29% ZnO. The accuracy of our combinatorial approach was 5%.

## 1. Introduction

The important applications for metal oxides, such as smart windows and displays, have been widely studied recently, with respect to their properties and electrochromic behavior. Electrochromic films have been used to decrease the extra absorption of heat in buildings, in applications such as smart windows, automobile sunroofs, energy-effective glazing and mirrors. The smart window structure contains some solid electrolyte and metal oxide as an electrochromic material layer sandwiched between transparent conductive layers. Recently, energy efficiency has been affected, and research has focused on energy-saving strategies in this important field.

The electrochromic process is based on a reversible redox process and characterized by coloration efficiency (CE). Energy-saving windows (smart windows) and energy storage systems, such as electrochromic (EC), photochromic and thermochromic systems, have been designed based on the same mechanism and both have sandwich device structures, and these are based on the electrochemical reaction of the electrode materials.

Transition metal (titanium, tungsten, nickel, vanadium, molybdenum and others) oxide films are the most interesting and most widely studied materials for this purpose. Conventional thin-film preparation methods include, for instance, chemical methods (spin coating, sol–gel deposition, chemical bath deposition, the Langmuir–Blodgett technique, etc.), chemical and physical vapor deposition and electrochemical methods (anodization, plating) (see ref. [1] and references therein).

Nevertheless, relatively few publications have studied the possible advantages (higher CE) of the mixtures of different metal oxides as electrochromic materials. The change in light absorption for the same electric charge represents the electrochromic effectiveness, and it can be higher in mixed metal oxide layers.

Earlier, we performed experiments with mixed metal oxides and found a positive effect on electrochromic behavior. Ismaeel et al. [2] determined the optimal composition of reactive magnetron-sputtered combinatorial mixed layers of titanium oxide and tin oxide (TiO_2_-SnO_2_) for electrochromic purposes. The maximum enhancement in light absorption was found for a (30%) TiO_2_–(70%) SnO_2_ composition.

In other experiments, a similar combinatorial material synthesis approach has been applied for a binary MoO_3_/WO_3_ system. We used organic propylene carbonate electrolyte cells in a conventional three-electrode configuration, and we carried out electrochromic redox reactions. We evaluated and plotted the CE against the composition values in the full range. The displayed data showed a characteristic maximum at around 60% MoO_3_. The accuracy of our combinatorial approach was 5% [3].

SE has been used by a lot of researchers as an investigation method for combinatorial or pure materials [4,5,6,7,8,9,10,11]. Fried et al. [12] used SE because it is a cost-effective, non-destructive and fast method for the mapping of WO_3_-MoO_3_ mixed layers. A thickness map and a composition map have been achieved by the developed optical models of the sample layers.

The objective of this work was to determine the CE and to investigate the electrochromic effectiveness of SnO_2_-ZnO mixed layers in a wide compositional range. A positive effect was achieved by using metal atoms with different diameters in the layers. ZnO was investigated for different purposes: Miccoli et al. [13] reported on the DC-sputtering deposition of ZnO:Al thin films as a transparent conductive oxide, and Semong et al. [14] synthesized gold-coated ZnO/Fe_3_O_4_ nanocomposites as a colorimetric-sensing detector.

There are different synthesis technologies for the fabrication of ZnO materials: carbo-thermal transport growth [15], electron beam evaporation [16] and in-plane surface epitaxy [17]. We chose the reactive magnetron-sputtering method, which is good for industrial (square meter)-scale covering purposes.

To the best of our knowledge, Zn has been investigated only as a dopant in other electrochromic metal oxides [18] while SnO_2_ or SnO_2_-metal oxide [19,20] mixtures have been studied only as photocatalytic materials. There are no publications where pure SnO_2_ or ZnO-SnO_2_ mixtures are studied as electrochromic materials.

In the frame of this article, we aim to study the electrochromic behavior of mixed zinc and tin oxides deposited using reactive DC magnetron sputtering. It should be noted that we apply a combinatorial approach for composition-graded layer deposition to allow study samples to be chosen from a full and continuous SnO_2(x)_ − ZnO_(1−x)_ composition range. We are not aware of any publications that have used a combinatorial approach in this field and with this material combination. The deposited films were characterized using spectroscopic ellipsometry (SE), Scanning Electron Microscopy (SEM) with Energy-Dispersive X-ray Spectroscopy (EDS) and coloration efficiency (CE) measurements. Our aim is to assess the results of investigations of these types of materials showing enhanced electrochromic behavior compared to pure materials. One can expect that mixing metal atoms with different sizes in films can enhance the CE.

## 2. Results

The main criterion for assessing EC device performance is the CE. Transmittance changes were measured in the straight-through direction during the coloration process, while the charge was calculated from the integral of the current vs. time data and the electrolyte-wetted area of the sample.

Figure 1a show an example normalized-intensity spectrum in the bleached state (the measured intensity divided by the intensity without the sample) of one sample point. One can see that below 400 nm (in the UV region), the layer is not fully transparent (the band gap energy of SnO_2_ is 3.6 eV (344 nm) and of ZnO is 3.3 eV (375 nm)). The behavior of the electrochromic materials is interesting in the visible region. Figure 1b shows curves of intensity versus time at five different wavelengths.

Figure 2a shows the calculated CE data as a function of the ZnO fraction of the film (different colors represent different wavelengths), while Figure 2b is a 3D representation of the same data. Individual points were calculated from the average of three independent measurements. The error is estimated to be 3%, calculated based on the accuracy of sample positioning in the measuring cell and the spot size of the optical beam.

The calculated CE data in Figure 2a and data values are given in Table 1 according to Equation (1).

A selection of the typical patterns of stable cyclic voltammetry (CV) (after several scans) is shown in Figure 3 using five different scanning rates (5, 10, 25, 50 and 100 mV/sec). The CV patterns show run-to-run stability after 2–3 scans and remain reproducible up to 10 scans.

The cathodic current follows a rising pattern, and a clear anodic peak is observable in all cases. It is important to note that an increased scan rate causes a twofold effect: (a) the observed anodic peak shifts towards negative values as the scan rate is increased, and (b) the anodic peak becomes more pronounced as the scan rate increases. Lower-scan-rate CV curves tend to form a symmetrical CV shape.

Garcia-Canadas and coworkers [21] found a similar CV pattern in the α-WO_3_ electrochromic oxide—LiClO_4_ electrolyte system. Their results show that a simple RC equivalent circuit allows us to explain the principal CV characteristics of lithium intercalation and deintercalation in amorphous films. Since our SnO_2(0.3)_ZnO_(0.7)_ film exhibits an analogous pattern, the anodic peaks in Figure 3 can be interpreted as a result of the series resistance effect.

## 3. Discussion

The CE maximum was found to be 29% Zn for each wavelength between 20 and 50 cm^2^/C. This 29% is very close to the optimum value of 30% in the case of the TiO_2_-SnO_2_ mixture which was investigated in our earlier paper [2]. We expected that mixing metal atoms with different diameters in the layers can enhance the CE. This (70–30)% mixture of different metal oxides seems to be the optimum for Li-diffusion in these sputtered materials.

The influence of mixing metal atoms with different diameters in the layers on EC behavior can be attributed to several factors. Mixing can create new pathways for charge carriers, enhancing the overall electrical conductivity. This increased conductivity can support faster ion intercalation and deintercalation processes, causing quicker color changes. Mixing can alter the electronic structure of the layer, affecting the way it absorbs and transmits light. These factors can explain the enhanced CE values.

## 4. Materials and Methods

The optimal composition of reactive magnetron-sputtered combinatorial mixed layers of tin oxide and zinc oxide (Sn_x_Zn_1−x_) was determined (where 0 < x < 1) for electrochromic purposes. Metallic Sn and Zn targets were placed separately from each other, and indium–tin-oxide (ITO)-covered glass and Si-probes on a glass substrate (30 cm × 30 cm) were moved under the two separated targets (Sn and Zn) in a reactive argon–oxygen (Ar-O_2_) gas mixture (see Figure 4a). The tin-zinc oxide layers were deposited onto ITO-covered 100 × 25 mm glass surfaces. Layer depositions were carried out by reactive sputtering in an (Ar + O_2_) gas mixture at a ~2 × 10^−4^ Pa base pressure and at a ~10^−1^ Pa process pressure. The target–substrate working distance was 6 cm. Volumetric flow rates of 30 sccm/s Ar and 70 sccm/s O_2_ were applied in the magnetron sputtering chamber. The plasma powers of the Sn and Zn metal targets were set to 800 and 1000 W, respectively. The samples were moved back and forth at a 25 cm/s walking speed between the Sn and Zn targets, and a mixed oxide film was deposited onto the ITO surface (see Figure 4b). A 5 min cooling interruption was applied after every 50 walking cycles.

X-ray Diffraction (XRD) measurements were performed on a Bruker AXS D8 Discover device (Billerica, MA, USA) to investigate the microstructure of the layers. We examined the Si-probes and show three characteristic diffractograms, one from the “Sn-side”, one from the “mixed-part” and one from the “Zn-side” and found that the “Sn-side” and the “mixed-part” layers are amorphous, but the “Zn-side” is a mixture of amorphous and nanocrystalline (with crystallites of less than 10 nm in size). Examples of the XRD measurements are shown in Figure 5.

Spectroscopic ellipsometry (SE) is an optical characterization technique with high accuracy [22]. We used the combinatorial approach to map our mixed metal oxides in the same way as in our earlier paper [12]. Different optical models, such as Effective Medium Approximation (EMA) and 2-Tauc-Lorentz Oscillator (2T–L), have were to achieve a composition map and a thickness map of the sample layers. We used SE in a similar manner to produce a composition map and a thickness map of our Sn-Zn combinatorial layers.

To determine the optimal composition for the best CE value, the layers were deposited onto ITO-covered glass. The composition map and thickness map were measured for the Si-probes (see Figure 4b). We checked the resulting compositional map for the Si-probes using an SCIOS2 Scanning Electron Microscope (SEM) (Thermo Fisher Scientific, Waltham, MA, USA) with Energy-Dispersive X-ray Spectroscopy (EDS) (see Figure 6) Zn-rich side: upper part, app. 50–50% is in the middle, Sn-rich side is below.

Figure 7 presents the comparison of the SEM and EDS measurements and shows a good match between them.

The CE η is given by the following equation:(1)$$\eta \left(\lambda \right)=\frac{\Delta OD\left(\lambda \right)}{q/A}=\frac{ln\;(\frac{T_{b}}{T_{c}})}{Q_{i}}$$
where *Q_i_* is the electrical charge inserted into the electrochromic material per unit area, ΔOD is the change in optical density, *T_b_* is the transmittance in the bleached state, and *T_c_* is the transmittance in the colored state. The unit of CE is cm^2^/C (square centimeters per Coulomb).

The CE was determined in a transmission electrochemical cell (see Figure 8). The cell was filled with 1 M lithium perchlorate (LiClO_4_)/propylene carbonate electrolyte. A 5 mm width masked (Sn-Zn oxide-free) ITO strip of the slides remained above the liquid level, allowing direct electric contact with the cell. A Pt wire counter electrode was placed into the electrolyte alongside a reference electrode. This arrangement was a fully functional electrochromic cell. The applied current was controlled through the cell using a Farnell U2722 Source Measurement Unit (SMU) (Leeds, UK). A constant current was applied through coloration and bleaching cycles of the electrochromic layer, and simultaneous spectral transmission measurements were performed by using a Woollam M2000 spectroscopic ellipsometer (Tokyo, Japan) in transmission mode.

The precision of the Sn/Zn ratio was 2%, while the precision of the position was 1 mm. Electrochemical measurements were carried out in a three-electrode cell arrangement where the potential was measured against the Ag/AgCl-KCl (3M) reference electrode. Samples of SnO_2_/ZnO mixed oxide films deposited onto ITO substrates were used as the working electrodes and were immersed into a solution of 0.1 M LiClO_4_ in propylene carbonate solvent. A platinum coil electrode was used as a reference. Cyclic voltammetry (CV) characterization was carried out using a Bio-Logic SP-50e potentiostat(Bio logic Systems Corp, Orlando, FL, USA) on a 1 × 1 cm sample with an oxide composition of SnO_2(0.3)_ZnO_(0.7)_ (i.e., at the maximum coloration efficiency).

## 5. Conclusions

We optimized the CE of a mixed tin oxide and Zn oxide (SnO_2_-ZnO) film deposited by reactive magnetron sputtering. We applied, for the first time, a combinatorial approach for composition-graded layer deposition to allow the study samples to be chosen from a full and continuous SnO_2(x)_ − ZnO_(1−x)_ composition range. We are not aware of any publications in this field that have used a combinatorial approach and with this material combination.

By using this combinatorial process, every composition (from 0 to 100%) was achieved in the same chamber after one sputtering process cycle. The mixed metal oxides showed at least 3 times better CE values than the pure oxides, because the electrochromic effectiveness can be higher in mixed metal-oxide layers and mixing metal atoms with different diameters in the layers can enhance the CE.

CE was the important parameter in this study. The maximum value of the CE was between 46 and 21 cm^2^/C between wavelengths of 400 and 800 nm at a ~71–29% Sn-Zn ratio. In the future, we need to optimize the optical modulation, the reversibility and the cycling stability with minimal performance degradation.

## Figures and Tables

**Figure 1 ijms-25-10801-f001:**
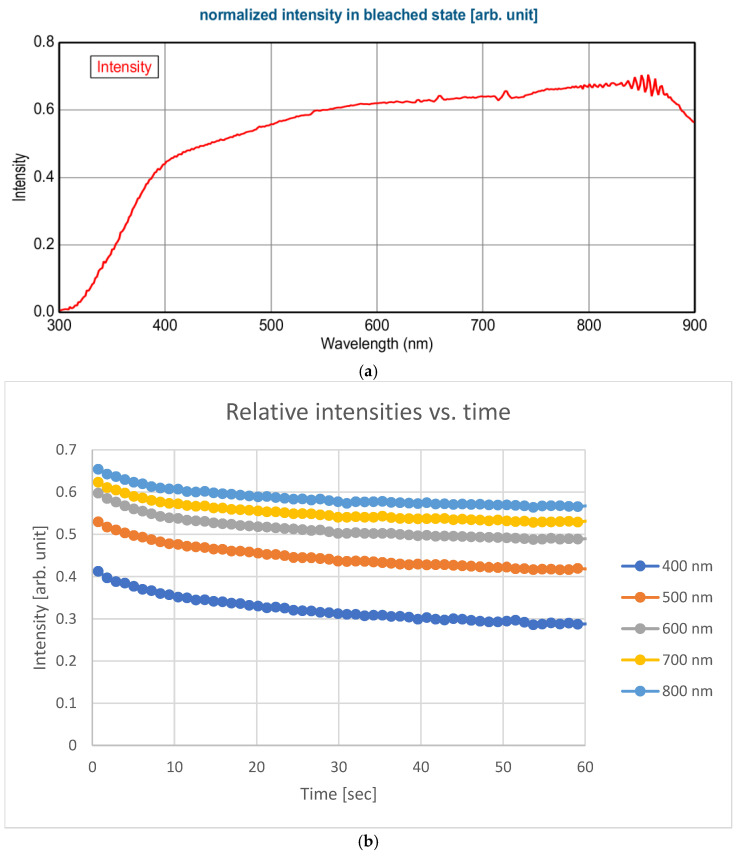
(**a**) Example normalized-intensity spectrum in bleached state (measured intensity divided by intensity without sample) of one sample point. (**b**) Curves of intensity versus time at five different wavelengths.

**Figure 2 ijms-25-10801-f002:**
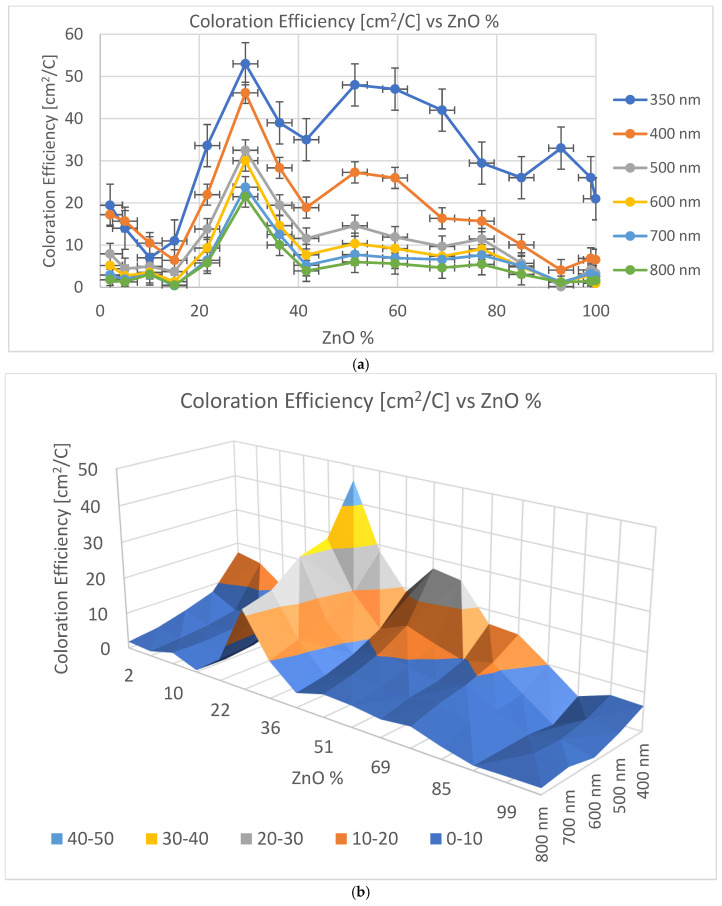
(**a**) CE of SnO_2_-ZnO vs. Zn % for wavelengths from 350 to 800 nm (individual color-coded curves represent different wavelengths: 1–350 nm, 2–400 nm, 3–500 nm, 4–600 nm, 5–700 nm, 6–800 nm. (**b**) Three-dimensional diagram of CE data of SnO_2_-ZnO vs. Zn % for wavelengths from 400 to 800 nm in visible spectral range.

**Figure 3 ijms-25-10801-f003:**
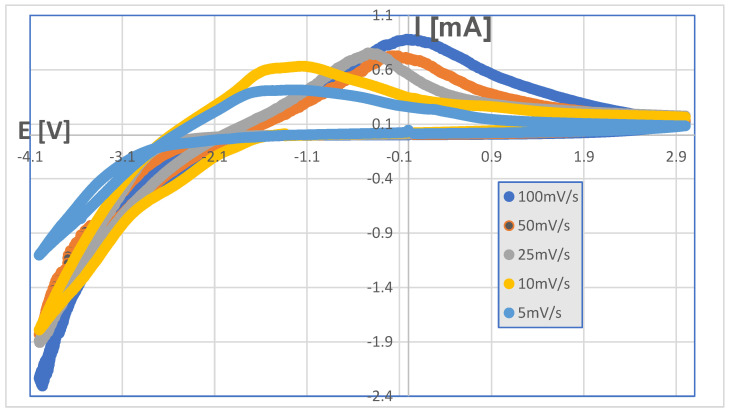
Cyclic voltammograms as a function of scanning rate taken on SnO_2(0.3)_ZnO_(0.7)_ oxide film in 1M LiClO_4_—propylene carbonate electrolyte.

**Figure 4 ijms-25-10801-f004:**
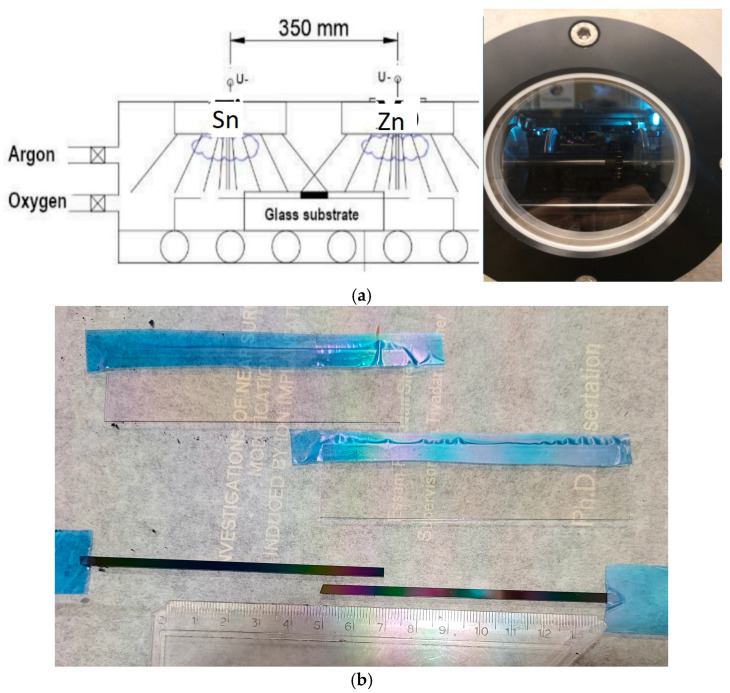
The SnO_2_-ZnO (**a**) arrangements of the two targets (at a 35 cm distance from each other) and the chamber for the DC magnetron sputtering after being vacuumed; the blue light is from the Ar-O_2_ plasma. (**b**) ITO-covered glass and Si-probes on a glass substrate before the electrochromic-experiments; the Sn on the left and the Zn on the right.

**Figure 5 ijms-25-10801-f005:**
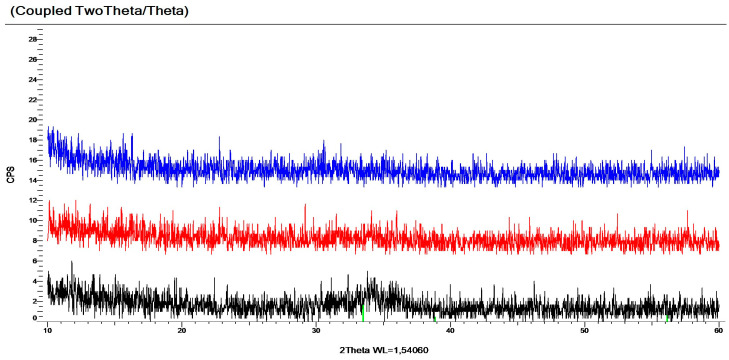
Examples of XRD measurements: one from the “Sn-side” (**upper**), one from the “mixed-part” (**middle**) and one from the “Zn-side” (**lower**). One can see only small and wide peaks (between 33 and 37 deg) in the lower diffractogram showing a trace of small ZnO nanocrystallites (with a less than 10 nm diameter).

**Figure 6 ijms-25-10801-f006:**
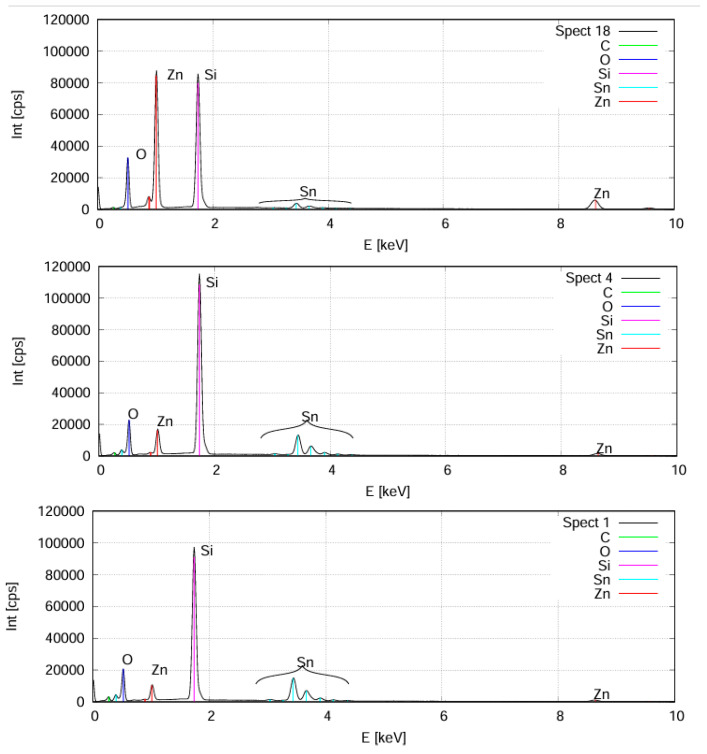
SEM-EDS spectra from the Si-probes: Zn-rich side (**upper**), app. 50–50% (**middle**), Sn-rich side (**below**).

**Figure 7 ijms-25-10801-f007:**
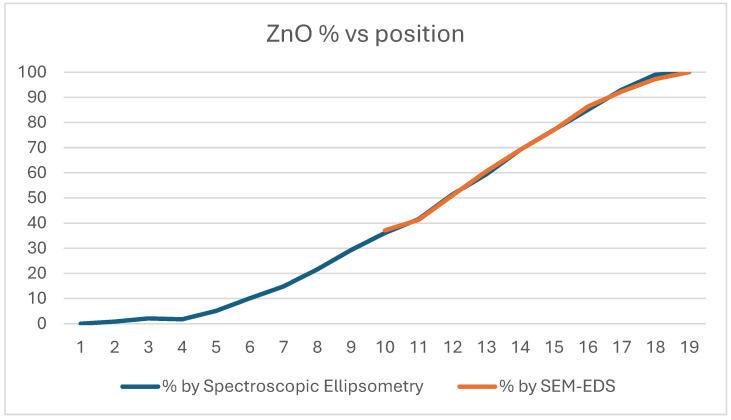
ZnO ratio measured on Si-probes by SE and SEM-EDS.

**Figure 8 ijms-25-10801-f008:**
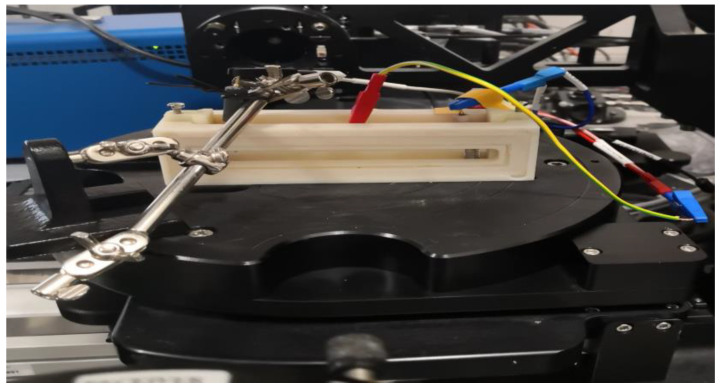
SnO_2_-ZnO during electrochromic experiments using SE.

**Table 1 ijms-25-10801-t001:** Calculated data for CE according to wavelengths of 400–800 nm. Unit of CE is cm^2^/C (square centimeters per Coulomb).

ZnO (%)	350 nm	400 nm	500 nm	600 nm	700 nm	800 nm
2	19.4	17.2	7.9	5.1	2.9	1.8
5	14.1	15.7	4.4	2.7	1.9	1.3
10	7.3	10.5	5.0	3.5	3.0	3.1
15	11.2	6.4	3.7	1.3	0.4	0.5
21.6	33.6	22.0	13.8	9.3	6.4	5.8
29.3	52.8	46.1	32.5	30.0	23.7	21.5
36.2	39.2	28.3	19.5	14.6	12.5	10.1
41.6	35.4	18.9	11.6	7.7	5.2	3.9
51.4	48.2	27.3	14.6	10.3	7.8	6.0
59.5	47.7	25.9	11.9	9.2	6.9	5.6
69	42.5	16.4	9.7	7.3	6.6	4.6
77	29.4	15.7	11.5	9.1	7.6	5.5
85	26.3	10.0	5.7	4.8	5.1	3.1
93	33.7	4.1	0.6	1.3	1.2	1.2
99	25.9	6.8	4.0	2.8	3.2	1.4
100	21.4	6.5	3.3	1.0	2.6	1.6

## Data Availability

The raw data supporting the conclusions of this article will be made available by the authors on request.

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
