# Peer review of "Compositional Optimization of Sputtered SnO2/ZnO Films for High Coloration Efficiency"

_ijms, 2024, doi:10.3390/ijms251910801_

Round 1

Reviewer 1 Report

Comments and Suggestions for Authors

This article studies the sputtered SnO2/ZnO films for high coloration efficiency.   

The manuscript needs to be revised and the following main points need to be carefully addressed before it can be considered for publication: 

1.     First, due to the large literature in the field, the novelty of the paper should be clearly addressed in the manuscript. 

2.     The manuscript presentation should be properly organized by appropriately moving the different sections. The paragraph “2. Result” should be called “Results” and moved after the section “Materials and Methods” so that the sequence and number of figures can be correctly reported. Therefore, the “Discussion” will be the last session before the Conclusions. 

3.     The introduction needs to be improved. For sake of completeness, among the other applications of ZnO-based materials, at least a relevant example of the important application of ZnO as transparent conductive oxide, as reported in this remarkable paper [https://doi.org/10.1016/j.apsusc.2014.05.225], and as a colorimetric-sensing detector [https://doi.org/10.1155/2023/3603680] should be mentioned. 

4.     Still in the introduction, it is necessary to briefly give to the readers a general overview of the different synthesis technologies, beyond sputtering, used in the literature for the fabrication of ZnO materials, by mentioning at least the carbo-thermal transport growth [DOI:10.1007/s00339-007-3946-4], the electron beam evaporation [https://doi.org/10.1016/j.apsusc.2008.09.026], and the in-plane surface epitaxy [https://doi.org/10.1021/acs.jpcc.7b02299]

5.     At line 91, regarding Fig.4 the MoO3 compound is mentioned, but it seems from Fig.4 that this compound is not involved in the presented results. This point must be properly clarified and corrected. 

6.     Analogously, in the caption of Fig. 5, Mo% is mentioned, but it seems that the Figure does not report any result on it. 

7.     The values reported in Table 2 are not clear and need to be described carefully in the text. 

8.     It is required that the scientific discussion of the results achieved be deeply commented insite the manuscript. 

9.     Furthermore, the conclusions should highlight the value that the manuscript adds to the current literature in the field and outline some perspectives. 

10.  The numbering of the references is wrong, the first and second reference being reported both as [1]. 

Comments on the Quality of English Language

The English needs to be improved. It is recommended that the manuscript be reviewed by a native English speaker. 

Author Response

This article studies the sputtered SnO2/ZnO films for high coloration efficiency.   

The manuscript needs to be revised and the following main points need to be carefully addressed before it can be considered for publication: 

  1. First, due to the large literature in the field, the novelty of the paper should be clearly addressed in the manuscript. 

We added the following text: ”In the frame of this article, we aim to study the electrochromic behavior of mixed Zinc and Tin oxides deposited using reactive DC magnetron sputtering. It should be noted that we apply a combinatorial approach for composition-graded layer deposition in order to allow study samples chosen from a full and continuous SnO2(x) − ZnO(1−x) composition range. We are not aware of any publications that have used a combinatorial approach in this field and with this material combination. The deposited films were characterized using spectroscopic ellipsometry (SE), Scanning Electron Microscopy (SEM) with Energy-Dispersive X-ray Spectroscopy (EDS), and Coloration Efficiency (CE) measurements. Our aim is to assess the results of investigations of this type of materials showing the enhanced electrochromic behavior compared to the pure materials. One can expect that mixing metal atoms with different sizes in the films can enhance the CE.

  1. The manuscript presentation should be properly organized by appropriately moving the different sections. The paragraph “2. Result” should be called “Results” and moved after the section “Materials and Methods” so that the sequence and number of figures can be correctly reported. Therefore, the “Discussion” will be the last session before the Conclusions. 

      We changed the order.

  1. The introduction needs to be improved. For sake of completeness, among the other applications of ZnO-based materials, at least a relevant example of the important application of ZnO as transparent conductive oxide, as reported in this remarkable paper [https://doi.org/10.1016/j.apsusc.2014.05.225], and as a colorimetric-sensing detector [https://doi.org/10.1155/2023/3603680] should be mentioned. 

We mentioned these publications.

  1. Still in the introduction, it is necessary to briefly give to the readers a general overview of the different synthesis technologies, beyond sputtering, used in the literature for the fabrication of ZnO materials, by mentioning at least the carbo-thermal transport growth [DOI:10.1007/s00339-007-3946-4], the electron beam evaporation [https://doi.org/10.1016/j.apsusc.2008.09.026], and the in-plane surface epitaxy [https://doi.org/10.1021/acs.jpcc.7b02299]. 

We involved these methods and citations into the Introduction.

  1. At line 91, regarding Fig.4 the MoO3 compound is mentioned, but it seems from Fig.4 that this compound is not involved in the presented results. This point must be properly clarified and corrected. 

We corrected it.

  1. Analogously, in the caption of Fig. 5, Mo% is mentioned, but it seems that the Figure does not report any result on it. 

We corrected it.

  1. The values reported in Table 2 are not clear and need to be described carefully in the text. 

We added the following text: “Figure 6 b shows the intensity curves versus time at five different wavelengths. The calculated CE data in Figure 6 c and data values are given in Table 1 according to equation (1).”

We changed the text of Table 1: Table 1. Calculated data for the CE according to the wavelengths 400-800 nm. The unit of CE is cm2/C (square centimeter per Coulomb).

  1. It is required that the scientific discussion of the results achieved be deeply commented insite the manuscript. 

We added the following text to the Discussion: “The influence of mixing metal atoms with different diameters in the layers on the EC behavior can be attributed to several factors. Mixing can create new pathways for charge carriers, enhancing the overall electrical conductivity. This increased conductivity can support faster ion intercalation and deintercalation processes, causing quicker color changes. Mixing can alter the electronic structure of the layer, affecting the way it absorbs and transmits light. These factors can explain the enhanced CE values.”

  1. Furthermore, the conclusions should highlight the value that the manuscript adds to the current literature in the field and outline some perspectives. 

We rewrote the Conclusion:

“We could optimize the CE of mixed Tin oxide and Zn oxide (SnO2-ZnO) layer deposited by reactive magnetron sputtering. We applied first time a combinatorial approach for composition-graded layer deposition in order to allow study samples chosen from a full and continuous SnO2(x) − ZnO(1−x) composition range. We are not aware of any publications that have used a combinatorial approach in this field and with this material combination.

By using this combinatorial process, every compositions (from 0 to 100%) were achieved in the same chamber after one sputtering. The mixed metal oxides showed at least 3 times better CE values than the pure oxides.

CE has been considered as the important parameter in this study. The maximum value of the CE is between 46 and 21 cm2/C between the wavelength 400 and 800 nm at ~ 71 % - 29 % Sn-Zn ratio. In the future, we need to optimize the optical modulation, the reversibility, and cycling stability with minimal performance degradation.”

  1. The numbering of the references is wrong, the first and second reference being reported both as [1]. 

Thank you, it was the mistake of Word, We corrected it.

Reviewer 2 Report

Comments and Suggestions for Authors

Comments to the Author
The authors of the manuscript ijms-3209326-peer-review-v1, reported deposition of mixed layers of Zinc oxide and Tin oxide (ZnO-SnO2) via magnetron sputtering and evaluate their Coloration Efficiency as a function of Zn % content. I think authors need to present more experimental data to support their claims. In the current version, authors presented the coloration efficiency as function of Zn% (Fig.4) and same results presented in Fig.5 as 3D graph and tableted in table 1.

-        Authors need to present the XRD, SEM or AFM, UV-transmission and Cyclic voltammogramms measurements before the manuscript to be accepted for publication.

-        The figures numbers need to be revised in the order of appearance on the manuscript

-        Page 2, line 91, authors stated “Figure 4 shows the calculated CE data as a function of MoO3 fraction”….I think the MoO3 need to revised to Zn %, as shown in Fig.4.

Author Response

Comments to the Author
The authors of the manuscript ijms-3209326-peer-review-v1, reported deposition of mixed layers of Zinc oxide and Tin oxide (ZnO-SnO2) via magnetron sputtering and evaluate their Coloration Efficiency as a function of Zn % content. I think authors need to present more experimental data to support their claims. In the current version, authors presented the coloration efficiency as function of Zn% (Fig.4) and same results presented in Fig.5 as 3D graph and tableted in table 1.

-        Authors need to present the XRD, SEM or AFM, UV-transmission and Cyclic voltammogramms measurements before the manuscript to be accepted for publication.

            We presented Figure 2 (XRD), Figure 3 (SEM-EDS), Figure 6 (relative Intensity, UV-transmission curve), and Figure 8 (Cyclic voltammogramms measurements)

-        The figures numbers need to be revised in the order of appearance on the manuscript

We revised the Figure numbers.

-        Page 2, line 91, authors stated “Figure 4 shows the calculated CE data as a function of MoO3 fraction”….I think the MoO3 need to revised to Zn %, as shown in Fig.4.

We corrected it.

Round 2

Reviewer 1 Report

Comments and Suggestions for Authors

The authors addressed all the raised issues, therefore the manuscript is now eligible for publication. 

Author Response

The authors addressed all the raised issues, therefore the manuscript is now eligible for publication.

Thank you your opinion and help!

Reviewer 2 Report

Comments and Suggestions for Authors

I have reviewed the revised ijms-3209326 manuscript. The authors did a real effort to improve the manuscript.  They added new measurements and revised some parts. However, the current version was hard to read and follow. Authors need to revise the figures numbers in accord with appearance in the manuscript. Remove all old parts and present clean version with final edition.

Author Response

I have reviewed the revised ijms-3209326 manuscript. The authors did a real effort to improve the manuscript.  They added new measurements and revised some parts. However, the current version was hard to read and follow. Authors need to revise the figures numbers in accord with appearance in the manuscript. Remove all old parts and present clean version with final edition.

Thank you the opinion and help. We have revised the Figure numbers and orders in accord with appearance.